# Managerial capacity among district health managers and its association with district performance: A comparative descriptive study of six districts in the Eastern Region of Ghana

**Anne Christine Stender Heerdegen**[1,2]*, **Moses Aikins**[3], **Samuel Amon**[3], **Samuel Agyei Agyemang**[3], **Kaspar Wyss**[1,2]

**1** Swiss Centre for International Health, Swiss Tropical and Public Health Institute, Basel, Switzerland, **2** University of Basel, Basel, Switzerland, **3** School of Public Health, College of Health Sciences, University of Ghana, Legon, Accra, Ghana

* annechristine.heerdegen@swisstph.ch

## Abstract

### Introduction

District health managers play a pivotal role in the delivery of basic health services in many countries, including Ghana, as they are responsible for converting inputs and resources such as, staff, supplies and equipment into effective services that are responsive to population needs. Weak management capacity among local health managers has been suggested as a major obstacle for responsive health service delivery. However, evidence on district health managers' competencies and its association with health system performance is scarce.

### Aim

To examine managerial capacity among district health managers and its association with health system performance in six districts in the Eastern Region of Ghana.

### Methods

Fifty-nine district health managers' in six different performing districts in the Eastern Region of Ghana completed a self-administered questionnaire measuring their management competencies and skills. In addition, the participants provided information on their socio-demographic background; previous management experience and training; the extent of available management support systems, and the dynamics within their district health management teams. A non-parametric one-way analysis was applied to test the association between management capacity and district performance, which was measured by 17 health indicators.

### Results

Shortcomings within different aspects of district management were identified, however there were no significant differences observed in the availability of support systems,

**Data Availability Statement:** Data cannot be shared publicly as it contains information that could compromise the privacy of research

participants. Data are available from the Department of Health Policy, Planning and Management,School of Public Health, College of Health Sciences, University of Ghana, Legon, Accra, Ghana (contact via Dr. Patrica Akweongo, Senior Lecturer/Head of Department akweongo@gmail.com).

**Funding:** This study is an integral part of the project PERFORM2Scale "Strengthening management at district level to support the achievement of Universal Health Coverage", which is funded by the European Commission's Seventh Framework program (FP7 Theme Health: H2020-EU.3.1.6., grant agreement number 733360).

**Competing interests:** The authors have declared that no competing interests exist.

characteristics and qualifications of district health managers across the different performing districts. Overall management capacity among district health managers were significantly higher in high performing districts compared with lower performing districts (p = 0.02). Furthermore, district health managers in better performing districts reported a higher extent of teamwork (p = 0.02), communication within their teams (p<0.01) and organizational commitment (p<0.01) compared with lower performing districts.

## Conclusion

The findings demonstrate individual and institutional capacity needs, and highlights the importance of developing management competencies and skills as well as positive team dynamics among health managers at district level.

## Introduction

Decentralization of health care, where authority and responsibilities for service delivery are transferred from higher levels (e.g. central, federal or national) to lower levels (e.g. state, regional, district, sub-district), is frequently perceived as a way to improve health system performance as local authorities are better able to make informed decisions regarding local conditions [1–3]. However, in order to improve performance, individual capacity among local health managers are needed [1, 4–6]. Moreover institutional capacities, such as functional support systems and enabling work environments, including an appropriate level of autonomy for the managers', must be in place [7].

This study focuses on district health managers (DHMs) working within District Health Management Teams (DHMTs) in Ghana. In Ghana, the DHMTs follow administrative directives issued by Ghana Health Service (GHS), the central level public health sector agency. The DHMTs have narrow decision-space with limited political and fiscal decentralization [8]. Nevertheless, they are mandated to convert inputs and resources, such as finance, staff, supplies, equipment and infrastructure into effective services that are responsive to the population needs [7, 9–11]. This mandate demands management capacity among the DHMs, defined as them having the abilities to keep the system functioning [12–14]; they must have the abilities to organize themselves effectively within the DHMTs in terms of encouraging teamwork, tackling problems collectively, spreading motivation and positive staff attitudes [15]. Moreover, they must have the abilities to manage health services (i.e. planning, supervising, monitoring quality and coverage), resources (i.e. staff, budgets, drugs, equipment, buildings and information), and stakeholders (external relations, partners, community members, service users and intersectoral stakeholders) [4, 16, 17]. Literature suggests weak management and leadership capacities among local health managers globally [7, 18–22]. However, a lack of tools for assessing management capacity among DHMs, results in limited knowledge about their actual competencies and qualifications [9, 18, 23].

Several studies suggest a positive association between district-level management capacity and health system performance [4, 24–30]. However, the study of Fetene et al in Ethiopia (2019) is to our knowledge the only in a lower income setting that has applied a quantifiable and precise measurement of management capacity at district level. Moreover, Fetene et al's study appear to be the first to investigate the association between district level management

capacity and health system performance measured by a wider set of public health indicators [4]. Further research is thus called for.

To enhance current knowledge on management capacity at district level and its association with health system performance in LMICs, this study aims to (1) explore qualifications and management competencies among DHMT members in Ghana, including characteristics of the DHMs and the systems they work in; and (2) to examine whether management capacity among DHMs is associated with health system performance.

Findings from this study can inform policy-makers and the global health community on areas in need of improvement for effective district health management in Ghana and other LMICs. Moreover, it can shed light on the importance of strengthening management capacity among local health managers in order to improve health system performance.

## Methods

### Ethics statement

This study was carried out as an integral part of the project PERFORM2Scale (P2S) under the lead of the Liverpool School of Tropical Medicine (LSTM). Ethical clearance was obtained from the Research Ethics Committee of LSTM (ID No.: 17–046) and the GHS Ethics Review Committee (ID No.: GHS-ERC:009/12/17). Additionally, permission was obtained from the Eastern Regional Health Administration. Written informed consent was obtained from all study participants and personal data were anonymized prior to analysis.

### Study design

This cross-sectional study, consisting of a self-administered survey, took place as part of the baseline assessment of the P2S project aiming to scale up a piloted district management strengthening intervention [31].

### Study setting

The survey took place in February and March 2018 in 6 of the 26 districts in the Eastern Region of Ghana. The study included the same districts as those in the P2S project. The districts were selected based on the following inclusion criteria: 1) willingness to participate in P2S, 2) them being clustered close to each other, 3) them representing different degrees of performance and (4) geographic entities (urban and rural). Characteristics of the study districts are available in Table 1.

### Study population

Inclusion criteria for participation in the study were 1) being employed as a DHMT member in one of the P2S study districts at the time of the study; and 2) having supervisory, administrative and management responsibilities within the study district.

### Data collection

Each DHM completed a self-administered questionnaire assessing their management competencies and skills, cf. S1 File. The questionnaire included 132 closed—and open-ended items divided into seven sections; 1) socio-demographic background; 2) management experience and exposure to management training; 3) functional management support structures and systems (i.e. for planning and budgeting, procurement of drugs and other commodities, data and human resource management); 4) general management competencies (i.e. interpersonal skills; leadership and conflict handling skills; time planning); 5) specific health system management

**Table 1. Characteristics of the six study districts.**

| | District 1 | District 2 | District 3 | District 4 | District 5 | District 6 |
|---|---|---|---|---|---|---|
| **Population** | 165.271 | 85.810 | 108.053 | 130.295 | 104.888 | 114.409 |
| **Sub-districts** | 7 | 7 | 9 | 7 | 6 | 7 |
| **Geographical setting** | Semi-Urban | Rural | Semi-Urban | Rural | Urban | Semi-urban |
| **Health Facilities** | 60 | 47 | 32 | 40 | 62 | 50 |
| District Hospitals | 1 | 1 | 1 | 1 | - | 2 |
| Health Centers | 6 | 5 | 5 | 2 | 13 | 9 |
| Maternity home | 2 | 2 | 0 | 2 | 3 | 2 |
| CHPS | 47 | 39 | 26 | 35 | 46 | 37 |
| **District Performance**[*] | 43.0 | 47.0 | 52.0 | 52.25 | 56.50 | 61.50 |
| **Burden of Disease**[**] | Malaria; Upper Respiratory Tract Infections;Anaemia | Malaria; Rheumatism and other joint pains; Upper Respiratory Tract Infections | Malaria; Diarrhea; Upper Respiratory Tract Infection | Malaria; Upper Respiratory Tract Infections; Rheumatism & other Joint pains | Malaria; Upper Respiratory Tract Infections; Rheumatism and other joint pains | Malaria; Upper Respiratory Tract Infections; Skin Diseases |

[*]Based on the League of District Performance,

[**]Based on OPD attendance

competencies (oversight and coordination; human resource management; resource management; financial management; information management; service delivery and community involvement); 6) overall management performance; and lastly 7) being part of a DHMT (team-work, communication, organizational commitment, job motivation and satisfaction among DHMs).

The DHMs rated their overall management capacity on a five-point Likert scale ranging from 1 (very poor) to 5 (excellent). The remaining items relating to competencies and being part of the DHMT were rated from 1 ("Strongly Disagree") to 5 ("Strongly agree"), while the availability of management support systems and structures were rated from 1 ("To a small extent") to 5 ("To a high extent").

A paper-based version of the questionnaire was distributed to DHMT members by members of the P2S research team. Prior to doing so, the aim of the survey was explained to the respondents and they were given the opportunity to ask questions for clarity.

### Validity of the questionnaire

The survey measuring managerial capacity among DHMs were developed by the authors due to absence of an existing assessment tool at the time of the study. The survey was developed based on a 1) literature review on what facilitates good management at district level in LMICs, and 2) on in-depth interviews conducted as part of the P2S initial context analysis, with DHMs, as well as with their supervisors (Regional Health Administrators) and peers (NGOs working within the study districts). To further ensure face and content validity, the questionnaire was developed and reviewed in an iterative process with five experts from the P2S consortium, including professionals from Ghana to ensure the appropriateness for a Ghanaian context. A total of 109 items were specifically developed for this study, while 22 were existing validated indexes.

The questionnaire was validated through five separate cognitive interviews with five DHMs' in two non-study districts in the Eastern Region, a similar approach to other studies

[32, 33]. The interviewees were asked to think loud when completing the questionnaire and explain why they responded as they did in order to identify questions that may elicit response error. The questionnaire was adjusted based on the five first interviews, and followed by five additional cognitive interviews with five other DHMs in the non-study districts.

## Data analysis

**District performance.** The dependent variable in our analysis is district performance. District performance is stated in Table 1, with District 1 having the lowest performance score and District 6 the highest performance score. Each of the study districts' performance was extracted from the Ghana League Table of District Performance (TDP), which includes data from the District Health Information Management System (DHIMS2) on 17 public health indicators, cf. Box 1, S2 File. The TDP ranks the 26 districts in the Eastern Region according to their aggregated annual performance score (average of four quarters). In 2017, the average annual score across the 26 districts ranged from 41.75 to 70.0 on a scale of 100.

**Independent variables.** Characteristics of the DHMs, i.e. their sex, age, educational background, previous management experience and training, as well as systemic factors, i.e. the number of DHMs in each DHMT and available management support systems and structures were included as independent variables.

The primary independent variable, namely the DHMs' management capacity, was measured by the item *"Overall, how would you rate your management and leadership skills and competencies?"*. In addition, sum variables measuring the DHMs' general management and leadership skills were included, i.e. their conflict handling and interpersonal skills (e.g. *"I ensure that staff under my supervision feel their contributions are valued and appreciated"*) (3 items), leadership skills (i.e. *"I am confident in my abilities to direct and motivate people I work with"*) (5 items), and time planning skills (*"I plan my workload by setting up daily/weekly/ monthly to-do-lists"*) (3 items). Moreover sum variables were included on the DHMs' competencies related to health system management, i.e. oversight and coordination (4 items), problem analysis (6 items), planning (7 items), implementation and monitoring (3 items) and reporting (2 items), as well as their skills within human resource management (11 items), resource management (4 items), information management (3 items), financial management (3 items) and service delivery and community involvement (3 items). The internal consistency of the sum scales were tested by the Cronbach alpha to ensure a reliability coefficient of 0.7 or higher [34, 35].

Lastly, validated scales were included measuring the DHMs' ability to organize themselves effectively within their DHMTs, i.e. their teamwork (7 items) [36], job satisfaction (6 items)

---

### Box 1: District performance health indicators included in the Ghana League Table of District Performance

**(1)** Outpatient Department Visits per capita, **(2)** Percentage of teenage pregnancies among ANC registrants, **(3)** Family planning Acceptor rate; **(4)** Percentage skilled deliveries, **(5)** Measles-Rubella-2 coverage, **(6)** Under 5 Malaria Case Fatality Rate, **(7)** % Pregnant women tested HIV positive, **(8)** Penta 3 coverage, **(9)** Isoniazid Preventive Therapy, **(10)** Antenatal Coverage, **(11)** Authorisation completeness, **(12)** Authorisation Timeliness, **(13)** Integrated Disease Surveillance and Response (IDSR) Weekly Timeliness, **(14)** IDSR Monthly Completeness, **(15)** Non Polio AFP rate, **(16)** Data Entry Completeness, **(17)** Data Entry Timeliness.

[37], motivation (3 items)[38] and organizational commitment in terms on their desire to stay (3 items) [37]. Moreover, an adapted and shortened version of Hoegl et al. items on communication within the DHMT (4 items) were included [39].

## Statistical analysis

Bivariate analyses were performed to evaluate differences across the different performing districts in DHM characteristics and available management support systems and structures. Nonparametric one-way analyses were applied, namely Fisher's exact test for categorical variables and Kruskal Wallis test for continuous variables.

In order to test the hypothesis that management capacity among DHMs was positively associated with health system performance, we used a non-parametric test for trend across ordered groups [40]. In light of the relatively low number of DHMs in each district, a multivariable regression model was not developed due to concerns about the reliability of the model.

All statistical analyses were performed with the statistical software Stata (Stata 14; StataCorp LP, College Station, TX, USA).

## Results

A total of 61 DHMs were invited to participate in the study. Hereof 59 completed the questionnaire (96.7% response rate). Non-respondents were caused by DHMs opting not to respond due to their busy schedule (n = 2). Six DHMs could not be included in the study as they were on maternity or sick leave (n = 4) or absent during the research teams' site visit (n = 2).

## Comparative analysis of district health managers' in different performing districts

Shortages of core administrative managers were observed in District 2, 4 and 6 (Table 2). All districts had a DHM within the technical areas of disease control, nutrition and health information, yet two districts were missing a Health Promotion Manager (4 and 5) and a Public Health Nurse (1 and 3). In terms of operational managers, half the districts were missing a HR Officer as well as a Supply Officer. Finance officers were present in all districts, except District 1.

There were no significant differences in the demographic and educational characteristics of the DHMs across the six districts (Table 3). Most of the DHMs' had a clinical background (20.3%) or a background in public health (35.4%). The most frequent highest educational qualification was a bachelor degree (44.1%) followed by a certificate/diploma (42.4%). More than one third of the respondents (34.5%) had less than 1 year of management experience prior to their current position. Moreover, less than half of the respondents (41.1%) had received formal training in management and leadership, i.e. degrees, certificate or diplomas. More than half (64.8%) had received informal management training within the last 12 months, i.e. mentoring, in-service training, non-certified programs.

## Differences in system factors across different performing districts

Table 4 demonstrates whether management structures and systems were in place to support DHMs in carrying out their role. There were no significant differences across the districts. All DHMs reported having received job descriptions specifying their respective tasks. However, the majority (88.1%) reported that they to a moderate/large extent took on additional roles and responsibilities that were not stated in their job description. All DHMs, except from in two districts (4 and 6), reported having access to relevant national and/or regional guidelines within the different work areas (i.e. on disease surveillance and response for disease control

**Table 2. Members of the district health management teams across study districts.**

|  | District 1 | District 2 | District 3 | District 4 | District 5 | District 6 |
|---|---|---|---|---|---|---|
| **Administrative managers** | **3** | **2** | **3** | **1** | **3** | **1** |
| *Director of Health Services* | ***1 | 1 | 1 | 1 | 1 | 1 |
| *Administrator* | 1 | 1 | 1 | - | 1 | - |
| *Dep. Dir. of Nursing Services* | 1 | - | 1 | - | 1 | - |
| **Technical managers** | **5** | **6** | **5** | **6** | **6** | **7** |
| *Public Health Nurse* | - | ×1 | - | 1 | 2 | *1 |
| *Disease Control Officer* | 2 | 2 | 2 | 2 | 2 | 2 |
| *Health Information Officer* | ***1 | 1 | 1 | 2 | 1 | 1 |
| *Nutrition Officer* | 1 | ×1 | 1 | 1 | 1 | 1 |
| *Health Promotion Officer* | 1 | 1 | 1 | - | - | **2 |
| **Operational managers** | **2** | **1** | **2** | **3** | **1** | **2** |
| Finance Officer | - | 1 | 1 | 1 | 1 | 1 |
| Supply/Procurement Officer | 1 | - | 1 | 1 | - | - |
| Human Resource Officer | 1 | - | - | 1 | - | 1 |
| **Other** | **2** | **1** | **1** | **-** | **3** | **1** |
| *Pharmacist* | 1 | - | - | - | - | - |
| *Mental Health Officer/Psychiatry Nurse* | - | 1 | 1 | - | **2 | - |
| *Principal Nursing Officer/CHN* | 1 | - | - | - | *1 | - |
| *Quality Assurance Officer* | - | - | - | - | - | *1 |
| **Total (n = 67)** | 12 | 10 | 11 | 10 | 13 | 11 |
| **Active DHMT members (n = 63)** | 12 | 10 | 11 | 10 | 11 | 9 |
| **Responded to survey (n = 59)** | 10 | 8 | 11 | 10 | 11 | 9 |

CHN: Community Health Nurse,

*Sick/maternity leave;

**1 missing due to sick leave;

***Absent;

×Non-respondents

officer, postings of health workers for HR officers, budgeting for finance officers). Regular team meetings (weekly) took place to a moderate (20.3%) or large extent (79.7%) in all districts, and records of team meetings were available (96.1%). In regards to supportive supervision, feedback and monitoring from supervisors, 10.3% reported receiving no or little supervision. Over half of the respondent reported inadequate funds (69.5%), logistics and infrastructure (55.9%) for carrying out their planned activities.

The majority of respondents rated support systems to be in place to a moderate or large extent, particularly in regards to data management (91.4%), procurement of drugs and other commodities (88.5%) and HR management (87.8%). The largest inadequacies were observed in terms of systems for planning and budgeting (19.6%), as well as for engaging communities (23.1%).

## The association between management capacity and district performance

As shown in Table 5, the DHMs overall rating of their management capacity was significantly associated with district performance; the self-assessed management capacity tended to increase from the low to high performing districts (p = 0.017).

The difference across district groups in general management and leadership competencies was not significant (Fig 1). However, as shown in Fig 1, there is an evident tendency among

**Table 3. Characteristics of study participants across different performing district.**

| | D1 (n = 10) | D2 (n = 8) | D3 (n = 11) | D4 (n = 10) | D5 (n = 11) | D6 (n = 9) | Total (n = 59) | *p |
|---|---|---|---|---|---|---|---|---|
| | n (%) | n (%) | n (%) | n (%) | n (%) | n (%) | n (%) | |
| **Sex** | | | | | | | | 0.98 |
| Male | 5 (50.0) | 5 (62.5) | 6 (54.5) | 5 (50.0) | 5 (45.5) | 4 (44.4) | 30 (50.9) | |
| Female | 5 (50.0) | 3 (37.5) | 5 (45.5) | 5 (50.0) | 6 (54.5) | 5 (55.6) | 29 (49.1) | |
| **Age** | | | | | | | | **0.37 |
| Mean (range) | 39.1 (29–57) | 39.5 (32–54) | 41.6 (30–58) | 35.6 (30–54) | 42 (33–57) | 36.8 (27–55) | 39.2 (27–58) | |
| **Educational background** | | | | | | | | 0.99 |
| Public Health | 3 (30.0) | 3 (37.5) | 5 (45.4) | 3 (30.0) | 3 (27.7) | 4 (44.4) | 21 (35.6) | |
| Medical Doctor /Nursing/Midwife | 2 (20.0) | 3 (37.5) | 1 (9.1) | 2 (20.0) | 3 (27.7) | 1 (11.1) | 12 (20.3) | |
| Accounting/Financing | 0 | 1 (12.5) | 1 (9.1) | 2 (20.0) | 1 (9.1) | 1 (11.1) | 6 (10.2) | |
| Human Resource Management | 1 (10.0) | 0 | 1 (9.1) | 1 (10.0) | 0 | 0 | 3 (5.1) | |
| Nutrition | 1 (10.0) | 0 | 1 (9.1) | 1 (10.0) | 1 (9.1) | 1 (11.1) | 5 (8.5) | |
| Other | 3 (30.0) | 1 (12.5) | 2 (18.2) | 1 (10.0) | 3 (27.3) | 2 (22.2) | 12 (20.3) | |
| **Highest educational qualification** | | | | | | | | 0.97 |
| Certificate/Diploma | 4 (40.0) | 4 (50.0) | 6 (54.6) | 3 (30.0) | 4 (36.4) | 4 (44.4) | 25 (42.4) | |
| Bachelor | 5 (50.0) | 2 (25.0) | 4 (36.4) | 5 (50.0) | 6 (54.5) | 4 (44.4) | 26 (44.1) | |
| Master/PhD | 1 (10.0) | 2 (25.0) | 1 (9.1) | 2 (20.0) | 1 (9.1) | 1 (11.1) | 8 (13.6) | |
| **Years in current position** | | | | | | | | 0.52 |
| <5 | 6 (60.0) | 3 (37.5) | 3 (27.3) | 7 (70.0) | 7 (63.6) | 4 (44.4) | 30 (50.9) | |
| 5–10 | 3 (30.0) | 4 (50.0) | 6 (54.6) | 2 (20.0) | 2 (18.2) | 5 (55.6) | 22 (37.3) | |
| >10 | 1 (10.0) | 1 (12.5) | 2 (18.2) | 1 (10.0) | 2 (18.2) | 0 | 7 (11.9) | |
| **Previous management experience** | | | | | | | | 0.88 |
| <1yrs | 5 (50.0) | 3 (37.5) | 3 (30.0) | 3 (30.0) | 2 (18.2) | 4 (44.4) | 20 (34.5) | |
| 1-5yrs | 2 (20.0) | 2 (25.0) | 2 (20.0) | 6 (60.0) | 5 (45.5) | 3 (33.3) | 20 (34.5) | |
| 5+ yrs | 3 (30.0) | 3 (37.5) | 5 (50.0) | 1 (10.0) | 4 (36.4) | 2 (22.2) | 18 (31.0) | |
| **Experience from other DHMTs** | | | | | | | | 0.48 |
| Experience | 4 (40.0) | 6 (75.0) | 6 (54.5) | 4 (40.0) | 4 (36.4) | 6 (66.7) | 30 (50.9) | |
| No experience | 6 (60.0) | 2 (25.0) | 5 (45.5) | 6 (60.0) | 7 (63.6) | 3 (33.3) | 29 (49.2) | |
| **Formal Management & Leadership training** | | | | | | | | 0.40 |
| Formal training | 4 (44.4) | 2 (28.6) | 4 (40.0) | 7 (70.0) | 4 (36.4) | 2 (22.2) | 23 (41.1) | |
| No formal training | 5 (55.6) | 5 (71.4) | 6 (60.0) | 3 (30.0) | 7 (63.6) | 7 (77.8) | 33 (58.9) | |
| **Informal Management and Leadership training within the 12 months** | | | | | | | | 0.44 |
| Informal training | 7 (77.8) | 6 (85.7) | 6 (54.5) | 7 (78.8) | 5 (45.5) | 5 (45.5) | 35 (64.8) | |
| No informal training | 2 (22.2) | 1 (14.3) | 5 (45.5) | 2 (22.2) | 6 (54.5) | 6 (54.5) | 19 (35.2) | |

*Fisher's exact,

**Kruskal Wallis test,

1) Master of Science in Pharmacy (n = 1), Bachelor in Health Service Administration (n = 1), Diplomas in Management, Health Promotion and Disease Control (n = 3),
2) Masters in Disease Control and Prevention (n = 1), a Bachelor in Health Administration (n = 1) and a Bachelor in Business Administration (n = 1), 3) Bachelor's in Health Management (n = 1), Masters in General Management (n = 1), Diploma in Purchasing and Supply (n = 1), Master of Philosophy in Leadership (n = 1)

DHMs in the lower performing districts to rate their interpersonal, leadership and conflict handling skills more negatively than the DHMs in the higher performing districts.

Competencies within the various health system management domains did not differ significantly between the districts (Table 5). District health managers' across all districts appeared to be confident in their skills related to reporting and carrying out situational and problem analyses, yet less confident in their planning, implementation and monitoring skills. Competencies

**Table 4. Functional support systems across different performing district.**

| | D1 (n = 10) | D2 (n = 8) | D3 (n = 11) | D4 (n = 10) | D5 (n = 11) | D6 (n = 9) | Total (n = 59) | p* |
|---|---|---|---|---|---|---|---|---|
| | n (%) | n (%) | n (%) | n (%) | n (%) | n (%) | n (%) | |
| **Additional responsibilities besides what is stated in job description** | | | | | | | | 0.59 |
| *Not at all/Small extent* | 2 (20) | 0 | 2 (18.2) | 0 | 2 (18.2) | 1 (11.8) | 7 (11.8) | |
| *To a moderate/large extent* | 8 (80) | 8 (100) | 9 (81.8) | 10 (100) | 9 (81.8) | 8 (88.9) | 52 (88.1) | |
| **Access to relevant national and/or regional guidelines within your work area** | | | | | | | | 0.18 |
| *Not at all/Small extent* | 2 (20) | 0 | 2 (18.2) | 0 | 0 | 0 | 4 (6.9) | |
| *To a moderate/large extent* | 8 (80) | 7 (100) | 9 (81.8) | 10 (100) | 11 (100) | 9 (100) | 54 (93.1) | |
| **Regular team meetings** | | | | | | | | 0.87 |
| *To a moderate extent* | 2 (20.0) | 2 (25.0) | 2 (18.2) | 2 (20.0) | 1 (9.1) | 3 (33.3) | 12 (20.3) | |
| *To a large extent* | 8 (80.0) | 6 (75.0) | 9 (81.8) | 8 (80.0) | 10 (90.9) | 6 (66.7) | 47 (79.7) | |
| **Available records of team meetings** | | | | | | | | 0.93 |
| *Not at all/Small extent* | 0 | 0 | 0 | 1 (10.0) | 1 (9.1) | 0 | 2 (3.4) | |
| *To a moderate/large extent* | 10 (100) | 8 (100) | 11 (100) | 9 (90.0) | 10 (90.9) | 9 (100) | 57 (96.1) | |
| **Supportive supervision, feedback and mentoring from your supervisor** | | | | | | | | 0.12 |
| *Not at all/Small extent* | 0 | 1 (12.5) | 2 (18.2) | 3 (30.0) | 0 | 0 | 6 (10.3) | |
| *To a moderate/large extent* | 10 (100) | 7 (87.5) | 9 (81.8) | 7 (70.0) | 11 (100) | 8 (100) | 52 (89.7) | |
| **Adequate funds to carry out planned activities** | | | | | | | | 0.59 |
| *Not at all/Small extent* | 7 (70.0) | 6 (75.0) | 9 (81.8) | 7 (70.0) | 5 (45.5) | 7 (77.8) | 41 (69.5) | |
| *To a moderate/large extent* | 3 (30.0) | 2 (25.0) | 2 (18.2) | 3 (30.0) | 6 (54.6) | 2 (22.2) | 18 (30.5) | |
| **Adequate logistics and infrastructure to carry out planned activities** | | | | | | | | 0.42 |
| *Not at all/Small extent* | 5 (50.0) | 4. (50.0) | 8 (72.7) | 5 (50.0) | 4 (36.4) | 7 (77.8) | 33 (55.9) | |
| *To a moderate/large extent* | 5 (50.0) | 4 (50.0) | 3 (27.3) | 5 (50.0) | 7 (63.6) | 2 (22.2) | 26 (44.1) | |
| ***Are there systems and structures in place to support within the following areas?*** | | | | | | | | |
| **Planning and budgeting** | | | | | | | | 0.89 |
| *Not at all/Small extent* | 2 (22.2) | 1 (9.1) | 3 (30.0) | 2 (18.3) | 2 (25.0) | 1 (14.3) | 11 (19.6) | |
| *To a moderate/large extent* | 7 (77.9) | 10 (90.9) | 7 (70.0) | 9 (81.8) | 6 (75.0) | 6 (85.7) | 45 (80.4) | |
| **Procurement of drugs and other commodities** | | | | | | | | 0.42 |
| *Not at all/Small extent* | 2 (25.0) | 2 (20.0) | 1 (14.3) | 0 | 1 (12.5) | 0 | 6 (11.5) | |
| *To a moderate/large extent* | 6 (75.0) | 8 (80.0) | 6 (85.7) | 9 (100) | 7 (87.5) | 10 (100) | 46 (88.5) | |
| **Data management** | | | | | | | | 0.68 |
| *Not at all/Small extent* | 0 | 0 | 1 (11.1) | 2 (18.2) | 1 (12.5) | 1 (10.0) | 5 (8.6) | |
| *To a moderate/large extent* | 9 (100.0) | 11 (100.0) | 8 (88.9) | 9 (81.8) | 7 (87.5) | 9 (90.0) | 53 (91.4) | |
| **Human resource management** | | | | | | | | 0.98 |
| *Not at all/Small extent* | 1 (12.5) | 1 (10.0) | 1 (12.5) | 2 (20.0) | 1 (12.5) | 0 | 6 (12.2) | |
| *To a moderate/large extent* | 7 (87.5) | 9 (90.0) | 7 (87.5) | 8 (80.0) | 7 (87.5) | 5 (100) | 43 (87.8) | |
| **Community-level structures or groups that enable community involvement** | | | | | | | | 0.14 |
| *Not at all/Small extent* | 3 (42.9) | 1 (10.0) | 1 (11.1) | 5 (50.0) | 1 (12.5) | 1 (12.5) | 12 (23.1) | |
| *To a moderate/large extent* | 4 (57.1) | 9 (90.0) | 8 (88.9) | 5 (50.0) | 7 (87.5) | 7 (87.5) | 40 (76.9) | |

*Fisher's exact t-test

**Table 5. Management capacity across different performing districts.**

| District | District 1 | District 2 | District 3 | District 4 | District 5 | District 6 | Total | p* |
|---|---|---|---|---|---|---|---|---|
| **Overall Management Capacity** | n = 10 | n = 8 | n = 11 | n = 10 | n = 11 | n = 9 | n = 59 | |
| *Mean (SD)* | 3.5 (1.0) | 3.9 (0.6) | 4.1 (0.3) | 4.2 (0.6) | 4.3 (0.5) | 4.1 (0.3) | 4.0 (0.6) | **0.02** |
| **Oversight and Coordination** | n = 10 | n = 8 | n = 11 | n = 10 | n = 11 | n = 9 | n = 59 | |
| Situational analysis[1] | 4.4 (0.6) | 4.3 (0.5) | 4.5 (0.5) | 4.5 (0.6) | 4.5 (0.6) | 4.4 (0.6) | 4.4 (0.5) | 0.47 |
| Problem analysis[2] | 4.1 (0.7) | 4.5 (0.5) | 4.4 (0.5) | 4.4 (0.5) | 4.4 (0.7) | 4.5 (0.6) | 4.4 (0.6) | 0.25 |
| Planning[3] | 4.0 (0.9) | 4.4 (0.7) | 4.0 (0.7) | 3.9 (0.8) | 4.2 (0.8) | 4.3 (0.7) | 4.1 (0.8) | 0.63 |
| Implementation and Monitoring[4] | 4.1 (0.7) | 4.5 (0.8) | 4.1 (0.7) | 4.4 (0.7) | 4.4 (0.7) | 4.3 (0.7) | 4.3 (0.7) | 0.78 |
| Reporting[5] | 4.6 (0.7) | 4.6 (0.6) | 4.5 (0.5) | 4.5 (0.7) | 4.7 (0.5) | 4.5 (0.5) | 4.6 (0.5) | 0.84 |
| **Human Resource Management[6]** | n = 7 | n = 7 | n = 9 | n = 9 | n = 10 | n = 6 | n = 48 | |
| *Mean (SD)* | 4.5 (0.5) | 4.2 (0.6) | 4.1 (0.6) | 3.9 (0.9) | 4.3 (1.0) | 4.2 (0.8) | 4.2 (0.7) | 0.91 |
| **Resource Management[7]** | n = 9 | n = 7 | n = 8 | n = 10 | n = 10 | n = 9 | n = 53 | |
| *Mean (SD)* | 4.1 (0.8) | 4.5 (0.4) | 4.3 (0.6) | 3.7 (1.2) | 4.5 (0.5) | 3.8 (0.6) | 4.1 (0.8) | 0.18 |
| **Financial Management[8]** | n = 8 | n = 8 | n = 10 | n = 9 | n = 10 | n = 8 | n = 53 | |
| *Mean (SD)* | 3.9 (0.6) | 4.2 (1.1) | 4.2 (0.6) | 3.5 (1.2) | 4.1 (1.3) | 3.8 (1.2) | 3.9 (1.0) | 0.74 |
| **Information Management[9]** | n = 9 | n = 8 | n = 10 | n = 9 | n = 9 | n = 8 | n = 54 | |
| *Mean (SD)* | 4.4 (0.9) | 4.6 (0.9) | 4.5 (0.5) | 4.6 (0.5) | 4.7 (0.4) | 4.2 (0.6) | 4.5 (0.7) | 0.22 |
| **Service Delivery & Community Involvement[10]** | n = 9 | n = 6 | n = 8 | n = 7 | n = 7 | n = 7 | n = 44 | |
| *Mean (SD)* | 4.4 (0.5) | 4.7 (0.5) | 4.5 (0.5) | 4.5 (0.5) | 4.2 (0.7) | 4.0 (0.6) | 4.4 (0.6) | 0.17 |
| **Dynamics within the DHMT** | n = 10 | n = 8 | n = 11 | n = 10 | n = 11 | n = 9 | n = 59 | |
| Teamwork | 3.8 (0.8) | 4.1 (0.7) | 4.4 (0.4) | 4.2 (0.6) | 4.3 (0.5) | 4.5 (0.4) | 4.2 (0.6) | **0.01** |
| Communication | 3.8 (0.7) | 4.1 (0.8) | 4.5 (0.5) | 4.0 (0.8) | 4.6 (0.5) | 4.7 (0.4) | 4.3 (0.7) | **<0.01** |
| Motivation | 3.1 (0.9) | 3.3 (0.7) | 3.9 (0.7) | 4.0 (0.7) | 3.3 (0.5) | 3.7 (0.7) | 3.6 (0.7) | 0.12 |
| Job satisfaction | 3.5 (0.4) | 3.4 (0.6) | 3.5 (0.4) | 3.5 (0.7) | 3.6 (0.5) | 3.8 (0.5) | 3.5 (0.5) | 0.11 |
| Organizational commitment | 3.7 (0.5) | 3.6 (0.9) | 4.5 (0.6) | 4.0 (1.1) | 4.3 (0.7) | 4.6 (0.8) | 4.1 (0.8) | **<0.01** |

*Oneway non-parametric test for trend (nptrend);

Sum-scales mean score:

[1]) 4 items, Cronbach alpha 0.76,

[2]) 6 items, Cronbach alpha 0.88,

[3]) 7 items, Cronbach alpha 0.92,

[4]) 3 items, Cronbach alpha 0.92,

[5]) 2 items, Cronbach alpha 0.83,

[6]) 11 items, Cronbach alpha 0.95

[7]) 4 items, Cronbach alpha 0.93

[8]) 3 items, Cronbach alpha 0.84

[9]) 3 items, Cronbach alpha 0.93

[10]) 3 items, Cronbach alpha 0.73

within management of human resources, resources and finances were rated less positively compared with information management, service delivery and community involvement.

Dynamics among the DHMs, i.e. teamwork, communication and organizational commitment, differed across the districts (Table 5), with a tendency of higher ratings in the higher performing districts. The highest performing district group scored best on all measures, except motivation where District 3 had a higher average score.

A total of 55.2% of the respondents stated that they did not reach their objectives set in their Annual Appraisal Form for the previous year, and 94.9% stated a desire to improve their management competencies within certain areas (results not shown). The areas mentioned included financial management (n = 7), information management (n = 5), HR management

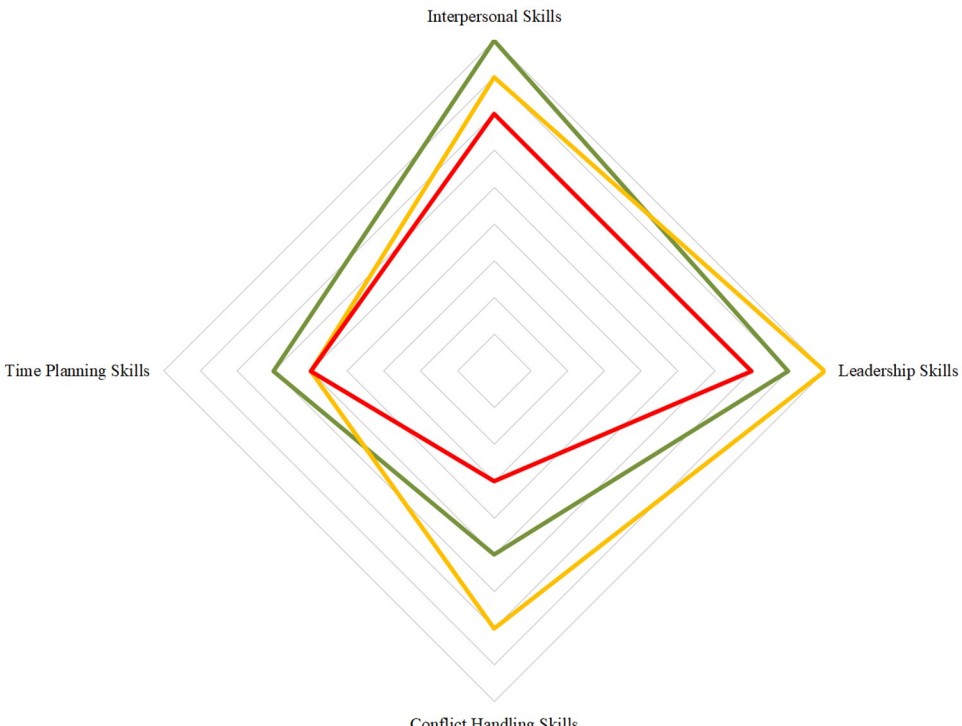

**Fig 1. General management and leadership competencies among district health managers in high, mid and low performing districts.** Red: Lowest performing districts (District 1 & 2), Yellow: Mid performing districts (District 3 & 4), Green: Highest performing districts (District 5 & 6). No statistical significant difference (p>0.05, oneway non-parametric test for trend); Interpersonal Skills: 2 items, Cronbach alpha 0.88, Time Planning Skills: 3 items, Cronbach alpha 0.86, Leadership Skills: 5 items, Cronbach alpha 0.9, Conflict Handling: 1 item.

(n = 2), general management and leadership skills (n = 12), conflict management (n = 4), time planning (n = 3), communication (n = 2), report writing (n = 3), how to involve community members (n = 2), advocacy and lobbying (n = 3) and supervision skills (n = 3). Lastly, seven people mentioned that they would like to improve specific competencies, i.e. research, reproductive and child health, malaria control and information management (i.e. gathering, analyzing and reporting information from facilities, stakeholders and communities to improve health services).

## Discussion

No systematic differences were observed across the different performing districts in terms of DHMT characteristics, functional support systems and specific management competencies. Nevertheless, differences were found in overall perceived management capacity, organizational commitment, teamwork and communication within the DHMTs. Management capacity and DHMT dynamics were positively correlated with health system performance.

### District health managers and the system they work in

Despite not being able to explain the differences in health system performance by systematic differences in the characteristics of DHMs and available functional management support systems, our findings contribute to health system research by identifying areas in which district health systems can be strengthened [9, 18, 23].

The WHO Leadership and Management Strengthening framework lists an adequate number of DHMs as an important element in having a robust health system [7]. We observed shortcomings in the number of core managers, converging with other studies in LMICs [7, 41, 42]. When staff shortages exist, current DHMs have to take on additional roles, which leaves them with a higher workload and tasks they do not have the appropriate competencies for. This will consequently affect their output [43], and may be the reason why more than half of the DHMs in this study did not reach their annual objectives. The DHMTs do not have authority to hire additional staff members, and thus depend on higher levels to recruit an adequate number of DHMs. This can be challenging in settings with a scare workforce, yet it is essential in order to have a well-functioning decentralized health system.

In our study, almost half of the DHMs had received formal management training. In Ghana, the District Directors of Health Services who lead the DHMTs, are strongly encouraged to undergo a certified management training course at the Ghana Institute of Management and Public Administration to prepare them for their role [44]. Thus, Ghana seems to be advancing in terms of competent DHMs compared to other countries, where DHMs often are described as clinical staff that have been promoted to management positions with little or no structured management training [12, 30, 43, 45, 46]. However, there is still improvement potential; more than one third of DHMs in this study had less than one year of management experience before entering their current role, and the majority had a bachelor degree or lower as highest educational qualification. Moreover, the majority of the respondents stated a desire to develop their competencies in order to carry out their job.

Additional areas with room for improvement include the extent of supervision provided to DHMs. Supervision plays a critical role in creating an enabling work environment where DHMs are appraised and incentivized to perform better [47–49]. Moreover, regular meetings and structures to enable community involvement should be present to a larger extent. Regular meetings have been described as an effective strategy to improve performance as they create a forum for addressing pressing issues, reflecting on progress and sharing information [6, 23, 43]. Community engagement can be helpful to identify needs and gain feedback from the community, in addition to enhancing accountability of health workers, and is thus an important component to train and empower DHMs in [46, 50].

This study demonstrates that DHMs perceive funds, logistics and infrastructure to be inadequate for carrying out planned activities. Irregularity in budgetary transfers in Ghana [50, 51], as well as in other countries [43, 52], results in DHMTs not having money to buy fuel and maintain vehicles. This affects their motivation negatively and prohibits them in carrying out essential supervision tasks [41, 51]. Apart from ensuring that resources are received by districts in a timely manner, DHMs' competencies in managing resources (financial, material and human) should be enhanced. Moreover, user-friendly systems for effective planning and budgeting should be in place.

## District management capacity and its association with health system performance

Overall, management competencies were positively rated by the DHMs in this study. However, potential for improvement was identified within certain areas, i.e. time planning, conflict handling and resource management (financial, material and human), converging with other studies on district health management [46, 49, 53]. Further research should look into the most effective ways to enhance such skills. Filerman suggest that essential management competencies are learned most effectively if the training takes place where the managers' work, within the team and it addresses existing challenges [22]. Gholipour et al describes that the academic credibility

of the instructors are important, and that meeting and sharing experiences with peers from other districts also can be an effective approach to strengthen district management [45].

A positive association was identified between the DHMs' overall management capacity and district health performance. Moreover, higher ratings of general management and leadership skills were observed among DHMs in the higher performing districts. The relatively small sample may have prevented the detection of a statistical significant association.

There was no association observed between district performance and the DHMs assessment of competencies within the specific health system domains. Moreover, the specific management competencies were generally rated more positively than the overall managerial capacity. The discrepancy may be caused by the fact that a subgroup of the DHMs responded to the items within the various domains, namely those involved in carrying out tasks related to the specific areas. These may thus have been particularly competent within their defined area of responsibility.

The higher ratings may also reflect areas, which frequently are targeted by management training, i.e. information management, reporting, implementation and monitoring. Technical skills relating to these areas are essential, yet do not necessarily improve the DHMs abilities to organize themselves effectively within the DHMTs, in terms of encouraging teamwork, tackling problems collectively, spreading motivation and positive staff attitudes. The DHMs assessments of communication, teamwork and organizational commitment were associated with their assessment of their overall managerial capacity, as well as with district performance. In alignment, Seims et al found that strengthening leadership and management skills among DHMs in Kenya through a team-based approach led to significant increases in health-service delivery [24]. The importance of building abilities among DHMs to work effectively within teams has been confirmed by other studies as well [5, 10, 23, 43, 54], and an emphasis on this should thus be ensured in district level management strengthening efforts.

It is important to note that management capacity among DHMs only is one of many factors that may affect district performance. Firstly, regardless of management competencies and skills, management practices may be constrained by the DHMs lack of authority to make decisions. Priorities at higher levels determine how DHMs carry out their responsibilities, which limits them in responding to the needs of their district [42, 51]. Secondly, district health system performance is affected by a myriad of other factors than district management, which were not adjusted for in current study. These include the economic status of a district, household economic conditions and poverty, general infrastructure, degree of urbanization, health facility management, social, religious affiliations and traditional beliefs that may render healthcare utilization [4, 55, 56]. Nevertheless, the positive association between district management and health system performance is confirmed in adjusted analyses performed in Fetene et al's study in Ethiopia. Their findings suggest that stronger management among district health officers magnify the positive effects of strong management at health facility level. Being the first point of primary care, management at health facilities play an important role in Ghana too [57], and future studies is suggested to explore management at health facility level and its synergy with management at district level.

## Measures of management capacity and health system performance

Management capacity is a complex concept to measure. This study measured it through a self-administered survey, which is an approach that has been used in other studies [45, 58–61]. The self-assessment methodology is founded on the concept of self-efficacy that posits that individuals who feels greater confidence in their ability to perform is more likely to successfully perform [30]. However, it is important to be aware of the fact that differences may exist between self-reported management behavior and actual behavior as observed by subordinates,

superiors and peers [62]. Managers' self-ratings tend to be inflated, which most likely also is the case in this study, where positive ratings in general were observed despite previous research suggesting a lack of capacity among DHMs. This study is to our knowledge the first study to measure management capacity through a questionnaire that were thoughtfully developed to assess management competencies among DHMs in a time and cost efficient way; validity and reliability was sought through expert review and cognitive interviews, which have been described to be effective when exploring new or poorly described concepts [32]. The identified correlation between self-reported managerial capacity and the objective measure of district performance, may be an indicator of criterion-related validity [63]. Nevertheless, further reliability and validity measures may enhance the effectiveness of the questionnaire; internal validity can be improved by including observations and additional assessments of the DHMs capacity from their superiors (Regional Health Administration), subordinates (health facilities) and peers (NGOs and other stakeholders) [64].

The outcome variable, district performance, is based on data from DHIMS2, and the analyses were thus carried out under the assumption that DHIMS2 data in Ghana is reliable [65]. However, it should be noted that inconsistencies can occur in DHMIS2, i.e. late or non-reporting from some health facilities, which could introduce misclassification bias of the exposure. Lastly, quality of care, which is an essential aspect of health system performance [66], was not reflected by the seventeen TDP indicators. Future research concerning health system performance should take this into consideration.

## Study limitations

The causal relationship between management capacity and performance cannot be established due to the cross-sectional study design; high performing districts may for instance have attracted highly competent DHMs or the confidence of DHMs self-assessment may have been affected by them being aware of the ranking of their districts' performance. If the district performance has affected the DHMs perception of their competencies, there may be differential misclassification and thus a biased measure of association. Moreover, the test of significance may have been influenced by the relatively small sample size; an effect that failed to be significant ($p < 0.05$) could prove significant in a larger samples. Further exploration of this research topic would benefit from longitudinal studies, as well as from larger studies with more statistical power and greater generalizability.

Due to the sample size, adjusted analyses could not be performed, and the confounding effect of mentioned factors affecting health system performance have not been established. Future studies may, if their sample size allows, eliminate confounding factors by running adjusted analyses, or by comparing performance between similar districts in regards to socio-demographic district characteristics. The latter was attempted in current study where there were no major differences across the districts in terms of DHM characteristics (Table 3) and availability of functional support systems (Table 4). Moreover, all study districts were located in the same region, governed by the same Regional Health Administration, and thus similar in terms of the DHMs level of authority, regional guidelines and procedures, climate and disease burden, cf. Table 1.

The inclusion of different performing, urban and rural districts makes current study findings generalizable to districts within the Eastern Region, yet to ensure external validity the questionnaire has to be tested in other settings.

## Conclusion

The complexity of district management and its association with health system performance is difficult to capture. However, despite the study limitations, our findings indicate a strong

association between self-reported management capacity and health system performance at district level in Ghana, which should be researched further. Moreover, this study identified areas within district health management that should be improved through policy making, i.e. inadequate supervision, funds and logistics available to DHMs, and targeted efforts, i.e. the DHMs motivation, specific management competencies and lack of management support systems.

## Supporting information

**S1 File. Questionnaire, district health managers' self-assessed management capacity.**
(DOCX)

**S2 File. District performance tables.**
(PDF)

## Acknowledgments

This study in an output from the PERFORM2Scale project (reference number: 733360): Strengthening management at district level to support the achievement of Universal Health Coverage, funded by the European Commission. The project involved a consortium of seven partners: Liverpool School of Tropical Medicine, Trinity College and Maynooth University, Ireland, Royal Tropical Institute, Amsterdam, School of Public Health, University of Ghana, Swiss Tropical and Public Health Institute, REACH Trust Malawi, School of Public Health, Makerere University.

Moreover, we would like to acknowledge and thank the members of the six District Health Management Teams who participated and took their precious time to speak with us and inform the findings of this study.

## Author Contributions

**Conceptualization:** Anne Christine Stender Heerdegen, Kaspar Wyss.

**Data curation:** Anne Christine Stender Heerdegen, Moses Aikins, Samuel Amon, Samuel Agyei Agyemang.

**Formal analysis:** Anne Christine Stender Heerdegen.

**Methodology:** Anne Christine Stender Heerdegen.

**Supervision:** Moses Aikins, Kaspar Wyss.

**Writing – original draft:** Anne Christine Stender Heerdegen.

**Writing – review & editing:** Moses Aikins, Samuel Amon, Samuel Agyei Agyemang, Kaspar Wyss.

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
