## [Decision Letter · Decision Letter 0]

3 Sep 2019

PONE-D-19-20019

District Health Management and its association with District Performance: A comparative descriptive study of six districts in the Eastern Region of Ghana

PLOS ONE

Dear Miss Heerdegen,

Thank you for submitting your manuscript to PLOS ONE. After careful consideration, we feel that it has merit but does not fully meet PLOS ONE’s publication criteria as it currently stands. Therefore, we invite you to submit a revised version of the manuscript that addresses the points raised during the review process.

The reviewers find the work of merit but have requested some additions and revisions, In addition to the items raised by the reviewers, please address the following points:

Clarify the concept of “District Health Management” in introduction and also modify it in method and title to avoid misunderstandings. In some case District Health Management means a structure of management and in your case it means the head managers of this structure not the management as a mechanism and structure.

Clarify whether you consider the role and capabilities of district health managers as district health management role and capabilities in discussion and limitation section.

You can find new references instead of old one. I recommend you discuss and compare your findings with the result of a series of paper we published in relation to district health management as a similar setting in a developing country. In this project we investigate the district health management environment in a developing country (Managerial barriers and challenges in Iran public health system: East Azerbaijan health managers' perspective. J Pak Med Assoc) after that develop management training program (Developing management capacity building package to district health manager in northwest of iran: a sequential mixed method study. J Pak Med Assoc), then design a management performance framework (A framework to assess management performance in district health systems: a qualitative and quantitative case study in Iran. Cadernos de saude publica) and finally implement and evaluate training programme (Evaluation of the district health management fellowship training programme: a case study in Iran. BMJ open).

Clarify the table topics,  such a “high”, “low”, “A”, “B”,… without any directive explanation.

Provide self-administered questionnaire, study protocol, informed consent as appendix.

Describe the self-administered questionnaire validation process in detail (how many expert, Validation methods, Analysis, …).

Describe cutoff point for Cronbach’s alpha coefficient and the result of your study.

Provide detailed information in relation to District Health Information Management System (DHIMS2) analysis method and process as appendix.

Consider confounding factor in your analysis and provide more detail in relation to controlling their in Statistical analysis section.

We would appreciate receiving your revised manuscript by Oct 18 2019 11:59PM. To enhance the reproducibility of your results, we recommend that if applicable you deposit your laboratory protocols in protocols.io, where a protocol can be assigned its own identifier (DOI) such that it can be cited independently in the future. For instructions see: http://journals.plos.org/plosone/s/submission-guidelines#loc-laboratory-protocols

We look forward to receiving your revised manuscript.

Kind regards,

Kamal Gholipour, PhD

Academic Editor

PLOS ONE

Journal Requirements:

Reviewers' comments:

Reviewer's Responses to Questions

**Comments to the Author**

1. Is the manuscript technically sound, and do the data support the conclusions?

Reviewer #1: Partly

Reviewer #2: Partly

Reviewer #3: Yes

Reviewer #4: No

2. Has the statistical analysis been performed appropriately and rigorously? 

Reviewer #1: Yes

Reviewer #2: No

Reviewer #3: Yes

Reviewer #4: No

3. Have the authors made all data underlying the findings in their manuscript fully available?

Reviewer #1: Yes

Reviewer #2: No

Reviewer #3: Yes

Reviewer #4: Yes

4. Is the manuscript presented in an intelligible fashion and written in standard English?

Reviewer #1: Yes

Reviewer #2: Yes

Reviewer #3: Yes

Reviewer #4: Yes

5. Review Comments to the Author

Reviewer #1: This is a descriptive study to assess the relationship between management capacity and district performance in the Eastern Region of Ghana. First, I would like to commend the authors for a comprehensive effort to assess public health/health management capacity in a LMIC context as well as for a well-written manuscript.

1. My primary concern relates to the generalizability of the study beyond those surveyed. The study was conducted in one region in the country. Further, in that region only 6 out 26 districts were assessed.

2. The authors conclude that there is an association between management capacity and district performance. While this is technically right (since they measured management capacity as perceived management capacity), all other objective measures of management competencies, included in this study, were not found to be associated with district performance. Thus, it is a bit hard for this reviewer to agree with the above-mentioned conclusion. Why was there a misalignment between the findings based on perceived management capacity and the objective measures of management competencies? Perhaps, (a) an association between self-reported capacity was observed primarily because the majority of the district managers in the low performing districts reported little prior management experience or training (as reported). Thus, these managers, because of the lack of training or experience viewed themselves as lacking in management skills, even though they may not have differed from other managers.

(b) There really was not enough variation in the performance measure to warrant the classification of three distinct groups (low medium, high). The cut-off points for these felt a bit arbitrary.

(c) Also it is possible that the lack of statistical significance across several of the indicators assessed may be due to a small sample size. I would love to see the authors discuss these above-mentioned issues a bit more.

3. Notably, what is a bit more apparent is the relationship between performance and work-place dynamics.

Minor

Abstract: Aim: May be missing "competencies" after "management"

Figure 1: Indicates (lines 234-237) that low performing districts have lower scores in time planning than mid-performing. The figure as shown indicates little difference between the two groups. Suggest that the authors exclude time planning from the list.

Reviewer #2: This is a study of the association between District Health Management and District Performance in six districts in the Eastern Region of Ghana. The benefit of this article is the deeper exploration than in previous studies, such as a self-administered questionnaire measuring the management capabilities and skills of district health managers. The downside is that this manuscript is very similar to the one by Fetene N et al (note the misspelling on line 74) who studied district-level health management and health system performance in Ethiopia.

Another problem is that, as the authors wrote in the study limitations, a causal relationship between the dependent variable and the independent variable cannot be established. The outcome variable, district performance, was not part of the questionnaire, rather it was extracted from aggregated data collected annually, so any association between the dependent and independent variables is questionable. Also, the authors did not investigate district health management implementers (such as doctors, nurses) and beneficiaries (such as patients), so the study was subjective and has no promotional value.

It is unclear how the authors actually assessed the outcome variables (high, medium, and low levels). In the methods section, the measurement standard cannot be found. What were the cut-points for high, medium and low and why were they chosen? In other words, the authors need to make a reasonable explanation for classifying the districts with scores of 61.5 and 56.5 into the high performance group, 52.3 and 52.0 into the middle performance group, and 47.0 and 43.0 into the low performance group.

The authors need to explain whether this is a random cluster sampling study or a convenience cluster sampling study. There is also no calculation of the sample size and exclusion criteria for the study in this manuscript.

Reviewer #3: The study is important in terms of management and health system performance in Ghana. The statistical analysis reflects the answers to the tasks set in the study. The author highlights the main problems but also reflects some limitations of the study which did not allow to fully elucidate the proposed goal.

Reviewer #4: I will suggest you revisit your statistical analysis, everything is wrong about it.

Why did you choose non-parametric test and not ANOVA parametric test?

Your data is two level hierarchical in nature where the 59 managers were nested in 6 district, you will need to account for this clustering in your analysis. SO therefore, I was thinking you should have applied some kind of multivariable logistic model while adjusting for clustering in 6 district.

Also, considering you categorized your outcome i.e. dependent variable into three level, you will need to applied multinomial logistic regression to this data.

That Table three is not right at all, you can not be applying kruskal-wallis test (for testing continuous variables) to categorical variables like sex, educational background, educational qualification, year in current position etc No you cant, this is slap on statistic face.

How on earth did you apply nptrend to ordered data when its not a time-series in nature? Why?

Your best bet:

Assuming you dont know how to do all I described above, please use ordinary CHI-SQUARE TEST, apply it to Table 3. Considering your sample size was low (n=59), please categorize, the performance outcome into two levels i.e. High vs mid/Low (combine Mid/Low together) then apply Chi-square test to EACH of the categorical variable. Then test Age (continuous variable using ANOVA) or you can categorize Age into 2 or 3 levels and then apply Chi-square test.

Then this will at least suffice.

But it will be better to apply Multivariable logistic regression/ Mixed effect logistic model to this data.

6. PLOS authors have the option to publish the peer review history of their article (what does this mean?). If published, this will include your full peer review and any attached files.

Reviewer #1: No

Reviewer #2: No

Reviewer #3: Yes: Ana Ciobanu

Reviewer #4: Yes: Dr. Babafela Awosile

---

## [Author Response · Author response to Decision Letter 0]

16 Oct 2019

The authors would like to sincerely thank the reviewers for their useful comments and suggestions, which enable us to improve the presentation of our study and the quality of our manuscript. Please find attached our point-by-point responses to the concerns raised by the reviewers and how we addressed these concerns.

---

## [Decision Letter · Decision Letter 1]

6 Dec 2019

PONE-D-19-20019R1

Managerial capacity among District Health Managers and its association with District Performance: A comparative descriptive study of six districts in the Eastern Region of Ghana

PLOS ONE

Dear Miss Heerdegen,

Thank you for submitting your manuscript to PLOS ONE. After careful consideration, we feel that it has merit but does not fully meet PLOS ONE’s publication criteria as it currently stands. Therefore, we invite you to submit a revised version of the manuscript that addresses the points raised during the review process.

Thank you for submitting your manuscript to PLOS ONE. After careful consideration, we feel that it has merit but does not fully meet PLOS ONE’s publication criteria as it currently stands.

1_ check that all statistic test values are included in your tables (e.g. chi square values etc.)

2_ The formatting in the reference list needs fixing -there is inconsistency in format of journal names (short or full) and in capitalization of titles

We would appreciate receiving your revised manuscript by Jan 20 2020 11:59PM. To enhance the reproducibility of your results, we recommend that if applicable you deposit your laboratory protocols in protocols.io, where a protocol can be assigned its own identifier (DOI) such that it can be cited independently in the future. For instructions see: http://journals.plos.org/plosone/s/submission-guidelines#loc-laboratory-protocols

We look forward to receiving your revised manuscript.

Kind regards,

Kamal Gholipour, PhD

Academic Editor

PLOS ONE

Reviewers' comments:

Reviewer's Responses to Questions

**Comments to the Author**

1. If the authors have adequately addressed your comments raised in a previous round of review and you feel that this manuscript is now acceptable for publication, you may indicate that here to bypass the “Comments to the Author” section, enter your conflict of interest statement in the “Confidential to Editor” section, and submit your "Accept" recommendation.

Reviewer #2: All comments have been addressed

Reviewer #4: All comments have been addressed

2. Is the manuscript technically sound, and do the data support the conclusions?

Reviewer #2: Yes

Reviewer #4: Yes

3. Has the statistical analysis been performed appropriately and rigorously? 

Reviewer #2: Yes

Reviewer #4: Yes

4. Have the authors made all data underlying the findings in their manuscript fully available?

Reviewer #2: Yes

Reviewer #4: Yes

5. Is the manuscript presented in an intelligible fashion and written in standard English?

Reviewer #2: Yes

Reviewer #4: Yes

6. Review Comments to the Author

Reviewer #2: Thank you for your answers and revisions. Personally, I think you have met the publishing requirements.

Reviewer #4: The review has improved the manuscript. I think the manuscript is better and the statistical analysis improved

7. PLOS authors have the option to publish the peer review history of their article (what does this mean?). If published, this will include your full peer review and any attached files.

Reviewer #2: No

Reviewer #4: No

---

## [Author Response · Author response to Decision Letter 1]

20 Dec 2019

The authors would like to sincerely thank the reviewers for taking the time to provide useful comments and suggestions. These have enabled us to improve the presentation of our study and the quality of our manuscript.

---

## [Editor Report · Decision Letter 2]

6 Jan 2020

Managerial capacity among District Health Managers and its association with District Performance: A comparative descriptive study of six districts in the Eastern Region of Ghana

PONE-D-19-20019R2

Dear Dr. Heerdegen,

We are pleased to inform you that your manuscript has been judged scientifically suitable for publication and will be formally accepted for publication once it complies with all outstanding technical requirements.

With kind regards,

Kamal Gholipour, PhD

Academic Editor

PLOS ONE
---

## [Editor Report · Acceptance letter]

13 Jan 2020

PONE-D-19-20019R2 

Managerial Capacity among District Health Managers and its Association with District Performance: A Comparative Descriptive Study of Six Districts in the Eastern Region of Ghana 

Dear Dr. Heerdegen:

I am pleased to inform you that your manuscript has been deemed suitable for publication in PLOS ONE. Congratulations! Your manuscript is now with our production department. 

With kind regards,

on behalf of

Dr. Kamal Gholipour 

Academic Editor

PLOS ONE